# Squeeze Film Damper Modeling: A Comprehensive Approach

**Edoardo Gheller \*** 🆔**, Steven Chatterton** 🆔**, Andrea Vania and Paolo Pennacchi** 🆔

Department of Mechanical Engineering, Politecnico di Milano, Via G. la Masa 1, 20156 Milan, Italy
\* Correspondence: edoardo.gheller@polimi.it

**Abstract:** Squeeze film dampers (SFDs) are components used in many industrial applications, ranging from turbochargers to jet engines. SFDs are applied when the vibration levels or some instability threatens the safe operation of the machine. However, modeling these components is difficult and somewhat counterintuitive due to the multiple complex phenomena involved. After a thorough investigation of the state of the art, the most relevant phenomena for the characterization of the SFDs are highlighted. Among them, oil film cavitation, air ingestion, and inertia are investigated and modeled. The paper then introduces a numerical model based on the Reynolds equation, discretized with the finite difference method. Different boundary conditions for oil feeding and discharging are implemented and investigated. The model is validated by means of experimental results available in the literature, whereas different designs and configurations of the feeding and sealing system are considered. Eventually, an example of the application of a SFD to a compressor rotor for the reduction of vibration and correction of the instability is proposed. The paper provides an insight regarding the critical aspects of modeling SFDs, underscoring the limits of the numerical model, and suggesting where to further develop and improve the modeling.

**Keywords:** squeeze film damper; seal instability; rotor dynamics; lubrication





## 1. Introduction

Vibrations represent an intrinsic problem in all fields of mechanical engineering including rotordynamics. Rotating machines are subject to remarkable loads and, with the development of machines that operate above some critical speeds, the control of vibrations is fundamental to guarantee long time operation. The typical problems in this field are excessive steady state synchronous vibration levels and subsynchronous rotor instabilities. The first one usually arises from excessive unbalance or due to operation close to a critical speed. The second one may depend on the presence of instability sources, connected to cross-coupling effects present in bearing systems and seals, among others. In some cases, the increase of the vibration, when crossing a critical speed during a runup or a rundown, can be harmful for the operation of the machine and the addition of some damping to the system is often required.

To this aim, squeeze film dampers remain one of the most effective components used because they offer the advantage of dissipating vibration energy when the shaft is supported by rolling element bearings. In addition, SFDs can improve the dynamic stability characteristics of rotor-bearing systems.

The most common design for these components is the one coupled with a rolling element bearing, as shown in Figure 1. The shaft is supported by a rolling element bearing and the coupling is often referenced as journal. The shaft vibration is transferred to the external ring of the bearing that "squeezes" the lubricant film, placed between the housing and the outer surface of the journal, generating high dynamic pressures. Therefore, dynamic forces counteract the lateral displacement of the shaft generating the damping effect. The anti-rotation pin is often applied to avoid any spinning motion of the journal, so that only translational displacements are possible, i.e., the journal can only translate or orbit without

spinning about its axis of symmetry. The shaft spinning is decoupled from the journal motion thanks to the presence of the bearing.

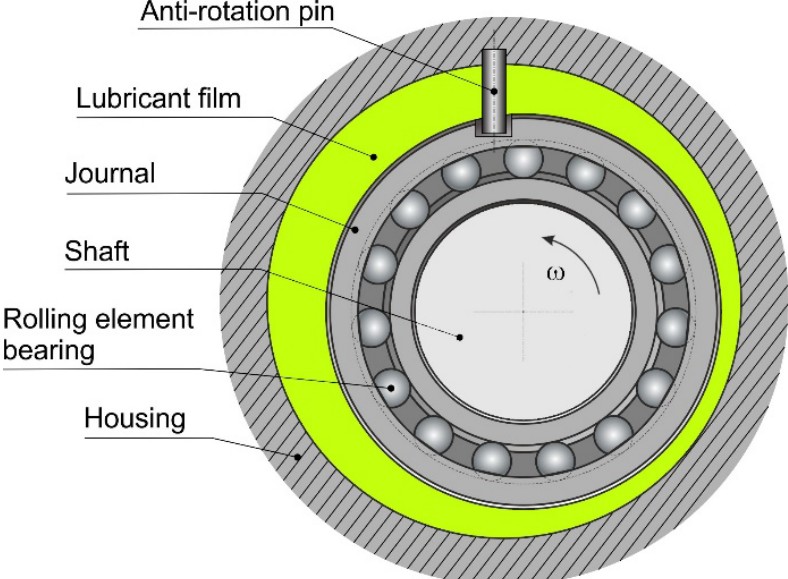

**Figure 1.** Simple squeeze film damper without centering mechanism, shaft rotation indicated by the arrow.

This configuration is characterized by strong non-linearities due to the "bottoming-out": the journal remains in contact with the casing surface at the run-up; when the level of the vibration is increased, the detachment of the two components happens resulting in a discontinuous change of the properties of the system. To reduce the non-linearity and the risk of collision between the static element and the whirling one in the case of large journal displacements, different supports are used, such as O-rings and squirrel cages. The selection of the proper stiffness of the support is fundamental for the correct operation of the SFD. If the support is too stiff, no relative motion between the shaft and the cage will be possible, i.e., no squeezing of the oil film; whereas if the stiffness is too low, the SFD can behave like a non-supported one [1,2].

Damping is the design parameter for all SFDs, and an optimal value for each application must be obtained. As a matter of fact, the utilization of a device whose damping capability is not aligned with the one requested by the system is useless if not dangerous. If the level of damping is too high, the SFD will dynamically behave as a rigid connection. Conversely, if the level of damping is too small, nothing will change in the dynamic response of the machine.

There are many studies in the literature that provide guidelines to determine the correct damping needed by a machine. In general, it depends on the dynamic characteristics of the machine itself, the typical operating conditions, and the kind of excitations [3,4].

Different models with different levels of complexity have been developed to predict the dynamic characteristics of SFDs. The first ones were based on the 1 D Reynolds equation for short plain journal bearings. This approximation is legitimate when the length to diameter ratio is lower than 0.25 and if no sealing mechanism is adopted, [5]. The effect of the SFD on the journal is modeled by means of linearized stiffness and damping coefficients likewise oil-film bearings. If no spinning motion is considered, no stiffening effect is obtained from the SFD. On the contrary, the long bearing approximation can be adopted when the length to diameter ratio tends to infinity or if seals limiting the oil flow are applied. In both cases, an analytical solution is possible. For this reason, many estimations of the coefficients are present in the literature [2,3].

The motion of the shaft is modelled, for convenience, as i) circular synchronous precessions, centered or with a static eccentricity, or ii) small amplitude motions about a

static displaced center. The first model is usually applied when the response to unbalance is investigated, the second one is used for critical speed and stability analyses as shown by San Andrés in [6].

From these early works, it is possible to understand that the clearance and the length to diameter ratio are two important parameters influencing the operation of the bearing together with the amplitude of the vibration, [1–3].

The 1-D Reynolds equation model has the advantage of simplicity, but the predictions can be considered reliable only for very simple geometries and for a limited range of operating conditions.

The main phenomena affecting the dynamic performance of SFDs are the fluid inertia, the liquid cavitation, the air ingestion, and the geometrical features.

Inertia is usually neglected in the derivation of the Reynolds equation, but for large clearances and amplitudes of motion, associated to higher vibrational frequencies, the added mass produced by the oil dynamic pressurization found experimentally has a value comparable to the mass of the entire SFD as highlighted by San Andrés and Vance in [7]. Different models that consider the effect of inertia can be found in the literature. As also reported by San Andrés and Vance in [8], for moderate values of the squeeze Reynolds number ($Re = \frac{\rho \omega c_l^2}{\mu} \leq 10$ with $\rho$ and $\mu$ being the density and dynamic viscosity of the oil respectively, $\omega$ the vibration frequency and $c_l$ the SFD clearance) the fluid inertia can be assumed not to affect the shape of the fluid purely viscous velocity profiles and consider the fluid temporal inertia in the modeling of SFDs. In [9], the effects of convective inertia and temporal inertia are considered together. In [10], a detailed description of the equations necessary to include the inertial contribution is presented together with an application.

Cavitation is stated as one of the principal reasons why predictions on the force coefficients, made with the simple model used in [1,5], do not fit the experimental results. For this reason, relevant effort has been put in the investigation and modeling of cavitation. In [11], Zeidan and Vance experimentally recognized five different cavitation regimes: un-cavitated film, cavitation bubble following the journal, oil–air mixture, vapor cavitation, vapor and gaseous cavitation. The second regime is considered as a transient condition, steady only for reduced whirling frequencies, that evolves in the third one with the shaft acceleration. The most common regimes are the third and fourth that sometimes combine with each other. Diaz and San Andrés in [12] concentrated mostly on vapor cavitation and air entrainment. They tested a bearing in open-ends and in fully flooded configuration, changing whirling frequencies and pressure of supply oil, and measuring the dynamic pressure generated. The authors showed the difference between the pressure evolution in time for the two-cavitation mechanism. For the vapor cavitation, the pressure profile is nearly identical for every cycle, while for air entrainment the pressure measurements showed great variability from one cycle to the other. Similar conclusions regarding the gaseous cavitation can be found in [13].

Due to the differences measured between the two phenomena, vapor cavitation and air ingestion are treated and modeled differently. Different vapor cavitation models and algorithms have been developed. The first cavitation model that was introduced is the so called $\pi$-film model, also known as Gumbel condition. Here, the relative pressure is considered zero in the region where it assumes negative values. According to this hypothesis, the ruptured film extends over half the angular length of the bearing. One of the most used algorithms is the so-called Elrod's cavitation algorithm, [14]. An evolution of this approach consists in the adoption of the linear complementarity problem (LCP), [15].

In [8,16,17] the effect of air ingestion and bubbly mixture is experimentally investigated. Air is "sucked" inside the SFD, and, after some cycles, the bubbles of air are finely dispersed in the mixture and persist also in the high-pressure zone. The presence of a compressible foamy mixture can explain the variability of the pressure's peak values. Different models that take into account the air ingestion are present in the literature. Among them, Diaz [18] provided a detailed procedure, supported by a series of experimental results, to include the air ingestion effect in the 2D Reynolds equation, based on the hypothesis

of a homogeneous bubbly mixture. To correctly determine the percentage of air inside of the mixture, a reference value is needed. In the experimental campaign, the air volume fraction is controlled at the feeding system. In industrial applications the SFD is fed with pure oil and air is ingested from the discharge locations. It is therefore necessary to predict the reference value of ingested air. In [19], the authors introduced a model to evaluate the air entrainment in open-ends short SFDs. Some years later, Mendez et al. [20] adapted Diaz's model to finite length bearings. Both the models presented in [18,20] are based on a simplified form of the Rayleigh–Plesset equation to model the presence of air bubbles in the oil in open-ends SFDs. Gehannin et al. in [21] considered instead the complete form of the equation and proposed a comparison with experimentally derived measures to evaluate the impact of these two different forms of the equation on the accuracy of the model.

Regarding the geometrical characteristics of the SFD, in [22], San Andrés et al. reported an extensive experimental campaign that thoroughly investigates the effect of different geometrical features on the dynamic properties of the SFDs. Six different configurations are tested, and the focus is set on the effect on the force coefficients of film clearance, length of the SFD, groove feeding and hole feeding, sealing ends and open ends, whirl orbit amplitude, shape of orbit, and number and disposition of feed holes.

In this paper, a comprehensive model based on the 2D Reynolds equation is introduced: The different phenomena described above are taken into considerations and discussed. The model is then validated with experimental and numerical data taken from the literature. The modeling of the different phenomena describing the dynamic behavior of SFDs is taken from several past works found in the literature. A simplified approach is considered to reduce the level of the difficulty and the parameters to be controlled. The goal of this work is to obtain a model that can be easily replicated and adapted.

In the literature there are more refined models based on the bulk-flow equations [23], and computational fluid dynamics [24–27]. Both approaches guarantee higher precision of the results, but the modeling and computational effort is higher than the one required by the model proposed in this work. The latter one gives the opportunity of investigating different phenomena in an approachable and straightforward way.

Eventually, an example of application of a SFD to a centrifugal compressor rotor for the reduction of vibration is proposed and a parametric investigation on the different parameters influencing the dynamic behavior of SFDs is performed. Moreover, the effect of the application of a SFD on the correction of an instability is also presented. In future works, the model proposed will be revised and improved to increase the accuracy.

## 2. Materials and Methods

The proposed model is based on the 2D Reynolds equation discretized with the finite difference approach. The inertia and air ingestion are modeled as extra terms of the Reynolds equation.

### 2.1. Oil Film Modeling

The approach to the analysis of the dynamic performance of SFDs is to simulate circular orbits of the shaft, whether centered (see Figure 2a) or not (Figure 2b), or small perturbations around the position of equilibrium. For simplicity, the proposed model is developed for centered circular orbits (CCOs), but it can easily be adopted for non-centered circular orbits or even noncircular orbits and oscillations around the equilibrium position if it is possible to identify a function that describes the behavior of the oil film thickness as a function of the time.

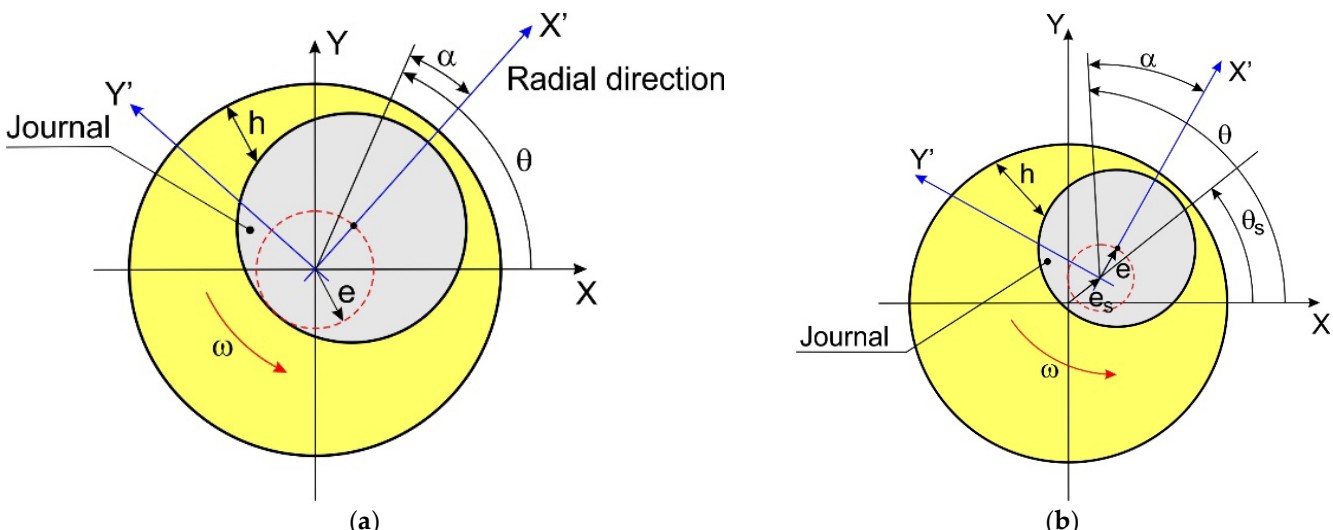

**Figure 2.** (**a**) Representation of circular centered orbit, (**b**) representation of circular eccentric orbit. Journal orbit represented by the red circle.

The rotation $\theta$ in the absolute frame of reference (X-Y) and the rotation $\alpha$ in the relative frame of reference (X'-Y') are related as follows:

$$\theta = \alpha + \omega t, \tag{1}$$

By considering the fixed reference system, it is possible to write the variation of the oil film thickness $h$ in time and space domains:

$$h(\theta, t) = c_l - (e \cos \omega t + e_s \cos \theta_s) \cos \theta - (e \sin \omega t + e_s \sin \theta_s) \sin \theta, \tag{2}$$

where $e$ is the orbit radius and, $e_s$ and $\theta_s$ are the amplitude and phase of the static eccentricity respectively (see Figure 2).

Considering Equation (1), at each time instant:

$$\frac{\partial}{\partial t} = -\omega \frac{\partial}{\partial \vartheta} = -\omega \frac{\partial}{\partial \theta}. \tag{3}$$

This assumption is valid only in case of CCOs and if the pressure field can be assumed to remain constant along the orbiting motion, no feedholes, discharge holes and piston ring seals. Moreover, if the orbiting frequency remains constant in time, Equation (3) allows to simplify every time derivative as a spatial one. This transformation allows to reduce the calculation time. In fact, it is not necessary to develop a time transient simulation since the simulation at one time instant is representative of the behavior of the oil for the entire orbit of the shaft.

## 2.2. Reynolds Equation

The general equations to describe the dynamic behavior of a viscous Newtonian fluid are the 3-D Navier–Stokes equations:

$$\frac{\partial \rho}{\partial t} + \nabla \cdot \left( \rho \vec{V} \right) = 0, \tag{4}$$

$$\rho \left( \frac{\partial \vec{V}}{\partial t} + \vec{V} \cdot \nabla \left( \vec{V} \right) \right) = -\nabla P + \nabla \cdot \left( \mu \nabla \vec{V} \right) + \nabla \left( -\frac{2\mu}{3} \nabla \cdot \vec{V} \right) + \rho g, \tag{5}$$

where (4) is the continuity equation and (5) are the conservation of momentum equations within the flow boundary.

Taking into consideration the SFD application it is possible to adopt some simplifying hypotheses, such as: (i) fluid density $\rho$ is considered constant, valid if cavitation is not present, (ii) fluid kinematic viscosity is constant, valid if temperature can be considered almost constant, (iii) inertia and body forces are neglected, (iv) fluid flow is considered laminar.

Finally, considering the SFD geometry the classical Reynolds equation can be obtained, [5]:

$$\frac{\partial}{\partial x}\left(h^3\frac{\partial P}{\partial x}\right) + \frac{\partial}{\partial y}\left(h^3\frac{\partial P}{\partial y}\right) = 12\mu\frac{\partial}{\partial t}(h). \tag{6}$$

In the case of constant whirling frequency, Equation (3) can be substituted inside Equation (6).

### 2.3. Fluid Inertia

In general, the fluid inertia forces are negligible if the value of the squeeze film Reynolds number is lower than 1. In case of high vibration frequencies, or SFDs with larger clearance, for example in case of inlet and outlet grooves, this value is greater than one and is usually lower than 50, [6]. As reported in [28,29], models that include inertia's effect give results closer to ones obtained experimentally for both force coefficients.

In this work, an approach similar to the one proposed in [30], a single Reynolds-like equation, is considered in which the effect of temporal inertia is added. Convective inertia terms are considered negligible as in [28]. Using cylindrical coordinates, the equation used in the model is:

$$\frac{\partial}{R\partial\theta}\left(\frac{h^3}{12\mu}\frac{\partial P}{R\partial\theta}\right) + \frac{\partial}{\partial y}\left(\frac{h^3}{12\mu}\frac{\partial P}{\partial y}\right) = \frac{\partial}{\partial t}(h) + \frac{\rho h^2}{12\mu}\frac{\partial^2 h}{\partial t^2}. \tag{7}$$

### 2.4. Air Ingestion

To fully consider the effect of air entrainment, the same approach adopted by the authors in [19] has been adopted. The Reynolds equation must be modified to consider the compressibility of the fluid and the effect of the presence of air bubbles on density and dynamic viscosity.

$$\frac{\partial}{R\partial\theta}\left(\frac{\rho h^3}{12\mu}\frac{\partial P}{R\partial\theta}\right) + \frac{\partial}{\partial y}\left(\frac{\rho h^3}{12\mu}\frac{\partial P}{\partial y}\right) = \frac{\partial}{\partial t}(\rho h) + \frac{\rho h^2}{12\mu}\frac{\partial^2 \rho h}{\partial t^2}, \tag{8}$$

$$\rho = (1-\beta)\rho_L, \tag{9}$$

$$\mu = (1-\beta)\mu_L, \tag{10}$$

$$\beta = \frac{1}{1 + \frac{P(x,t)-P_v}{P_{G\sigma}}\left(\frac{1}{\beta_0}-1\right)}. \tag{11}$$

where $\beta$ is the air–mixture volume fraction, $\beta_0$ is the reference value for $\beta$, $P_{G\sigma}$ is the pressure of the air bubble for the critical radius, $P_v$ is the vapor cavitation pressure, and $\mu_L$ and $\rho_L$ are the dynamic viscosity and density for the pure oil.

In [19], a model to evaluate $\beta_0$ is presented for short SFDs. However, the short-length bearing approximation is not always applicable. For example, it is limited to $L/D < 0.25$. In [20], it is proposed to numerically evaluate the volumetric inflow of air at the sides of finite length SFD and evaluate the new reference value of volume air fraction. The pressure cycle is then repeated with the updated value of $\beta_0$. This procedure is continued until the convergence on $\beta_0$ is reached. A procedure like this one has been adopted in the present work and the starting value of reference volume fraction of ingested air will be considered zero.

### 2.5. Negative Pressure Zone

As previously mentioned, different models and algorithms that deal with vapor cavitation have been adopted in the literature. In this work, the approach presented by Fan and Behdinan in [31,32] is applied. The vapor cavitation is solved as a linear complementarity problem as suggested in [15]. The algorithm proposed in [33] is applied in the solution of the *LCP*.

### 2.6. Geometrical Discretization

The cylindrical geometry of the SFD il flattened in a 2D plane, as shown in Figure 3.

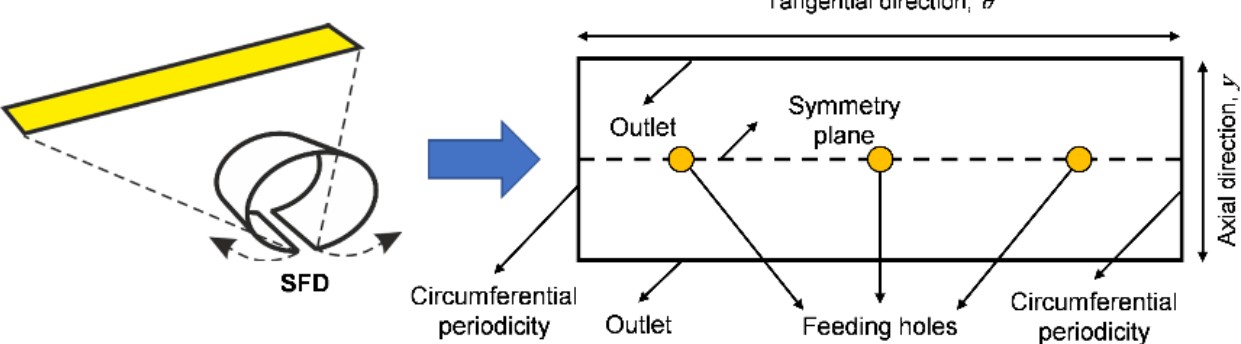

**Figure 3.** 2D geometrical transformation of the SFD.

A structured mesh is considered for the spatial discretization. The approach followed for the discretization of the holes is explained in the following sub-section.

### 2.7. Inlet Boundary Conditions

Different types of boundary conditions can be adopted at the inlet port. If the feeding holes are not considered in the modeling, it is possible to consider only half of the SFD by the symmetry boundary condition:

$$\left.\frac{\partial P}{\partial y}\right|_{Symmetry\ plane} = 0. \tag{12}$$

Conversely, if the feeding system is considered, the flowrate is imposed at each hole. If the laminar flow is assumed and no central groove is present, the flow rate is as follows:

$$Q_{inlet} = C_i\left(P_{supply} - P(x_h, z_h)\right)\left[\frac{m^3}{s}\right], \tag{13}$$

where $P(x_h, y_h)$ is the pressure of the oil at the hole location and $C_i$ is a coefficient that includes the orifice area and flow coefficient. In a 3-D model, this flow rate would be directed radially but, since this model is planar, it will be considered in the axial and tangential directions. A more detailed description can be found in [34].

The circular geometry of the hole is simplified as a rectangle, as shown in Figure 4. At the edges of the boundary, the pressure is constant and equal to the feeding one imposed at the center. Considering that the flow from the hole is delivered in both axial and tangential directions, the whole geometry of the SFD must be considered.

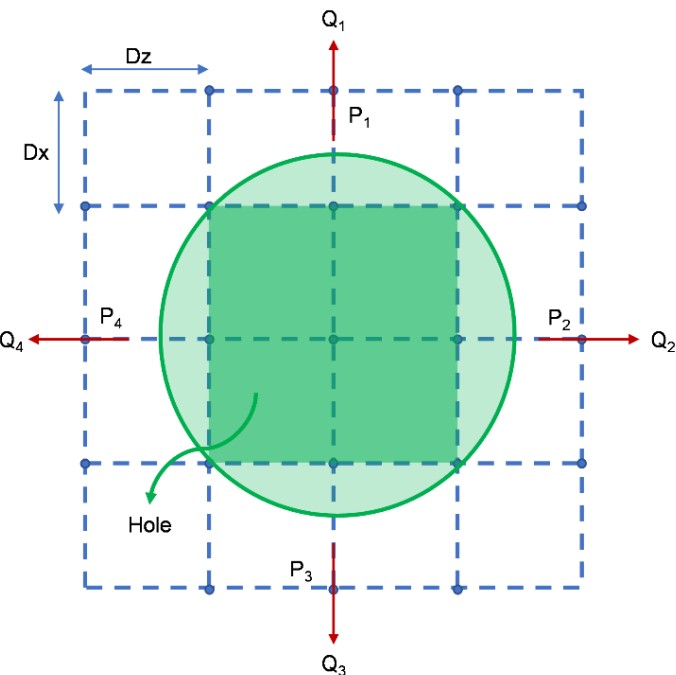

**Figure 4.** Mesh representation for the feeding hole boundary condition.

Considering the flow entering from one side of the discretized hole:

$$q_z = -\frac{h^3}{12\mu}\frac{\partial P}{\partial z}\left[\frac{m^2}{s}\right],\tag{14}$$

It is possible to solve

$$Q_1 = -\frac{\overline{h_1}^3}{12\mu}\left(\frac{P_1 - P_{supply}}{Dx}\right)2Dz,\tag{15}$$

where $P_1$ is the interpolation of the three points above the boundary of the hole, and $\overline{h_1}$ is the oil film height at $\frac{Dx}{2}$ from the side of the rectangle. Similar expressions can be written for the rest of the flow vectors. The same holds for all other sides $P_2$, $P_3$, and $P_4$.

In general, when the pressure of the oil inside the SFD in the vicinity of the feeding hole is higher than the supply pressure, a backflow happens: a flow rate of oil exits the land of the SFD and enters the supply circuit. As reported in [35], in practical application, check valves are applied to the feeding ducts to avoid backflows and to reduce the effect of pulsating pressure in the supply circuit. A detailed description of the application of check valve is present in [36]. For this reason, when the presence of feeding orifices is simulated in this model, Equation (13) will be used at the nodes where the orifices are located. If the pressure at the hole location is higher than the supply pressure, no boundary condition will be assigned.

In many applications, central grooves are applied as shown in Figure 5a.

In [34,37] it is proposed to model the feeding groove as a reservoir of oil at the feeding pressure. In [22], San Andrés et al. report instead that large values of dynamic pressure in the groove region occurred proving the previous assumption to be wrong. The same approach introduced in [28] is considered when modeling the presence of grooves. As shown in Figure 5b, the flow inside the groove is divided into two regions: a recirculating one and a through-flow close to the journal. Only the second one is active in the dynamic pressure generation, and therefore an effective groove depth is considered. Moreover, the feeding orifices are considered.

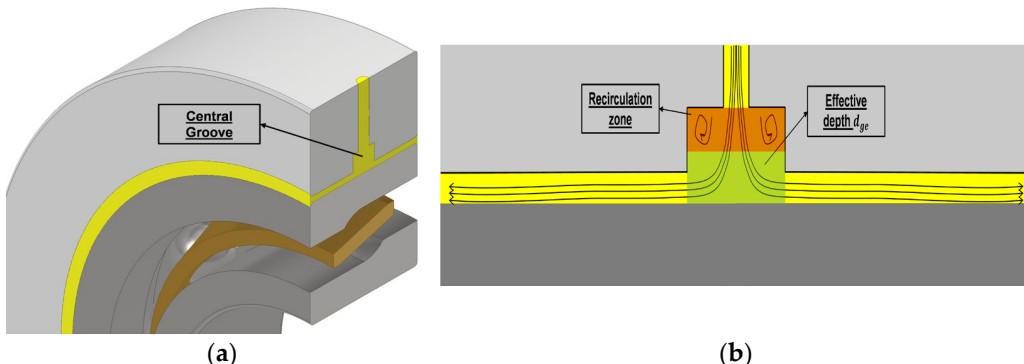

<center>(<b>a</b>)　　　　　　　　　　　　　　　　　　(<b>b</b>)</center>

**Figure 5.** 3D representation of SFD equipped with central groove (**a**); Schematic representation of the flow paths passing through the central groove (**b**). Adapted from [28].

The value of the effective groove depth ($d_{ge}$) is usually optimized using as benchmark the force coefficients obtained experimentally [28].

### 2.8. Outlet Boundary Condition

As reported in [5], many different boundary conditions can be assigned for the outlet section. In general, the SFD can be exposed to ambient pressure air, and in this case the boundary condition to be assigned is:

$$P(L,t) = P_{air}, \tag{16}$$

where $P_{air}$ is the ambient pressure at the outlet.

In this case the SFD is subjected to high air entrainment, comporting a reduction of the damping capacity of the device. With an open-ends configuration, the exiting flow rate is higher, a condition that will require a higher inlet flow rate of oil. For this reason, SFDs are usually sealed at the ends. The sealing is usually not complete otherwise. Due to the oil heating, the damping capacity would decrease. In the scientific literature, it is possible to find many types of sealing to reduce the leakage of the SFD. One of the most common is the piston ring shown in Figure 6 [5,35,38].

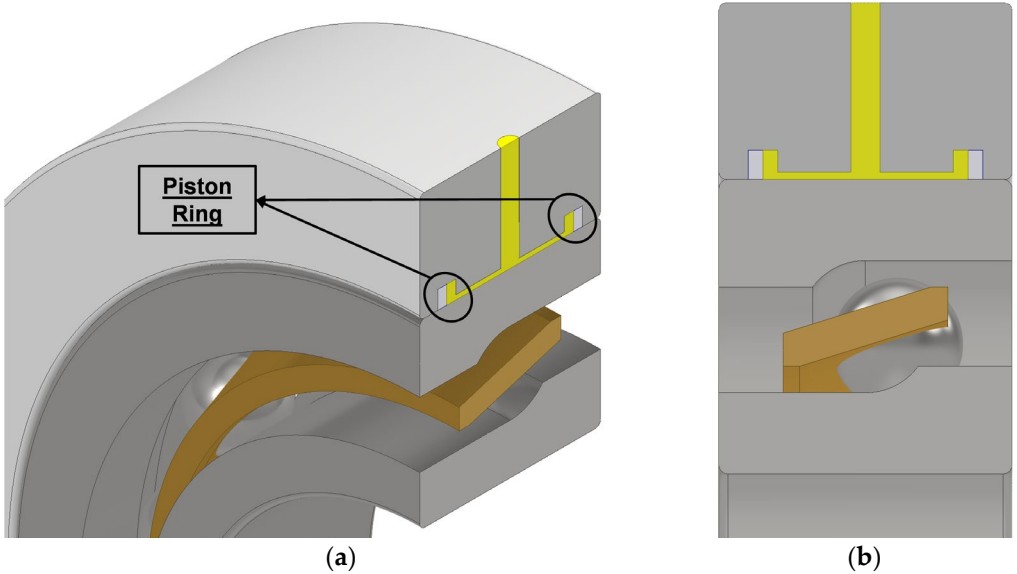

<center>(<b>a</b>)　　　　　　　　　　　　　　　　　　(<b>b</b>)</center>

**Figure 6.** 3D representation of SFD equipped with piston-ring as sealing mechanism at the outlet (**a**); Frontal view of SFD equipped with piston-ring as sealing mechanism at the outlet (**b**).

The piston ring seal represents a limitation on the outlet flowrate, and it can be defined as follows [34]:

$$q_{out} = \frac{C_p(P(\theta, L) - P_{out})h_p^3}{12\mu w_p} \left[\frac{m^2}{s}\right],$$ (17)

where $C_p$ is the piston ring loss coefficient, $0 < C_p < 1$, $P_{out}$ is the pressure outside the seal, usually ambient pressure, $h_p$ and $w_p$ are the piston ring radial gap and axial dimension respectively. $C_p$ has a main impact on the evaluation of the outlet flowrate. Moreover, it may not be straightforward to be obtained. In this application is considered as a tuning parameter to evaluate the effect of the seals.

Substituting Equation (17) into Equation (15):

$$\frac{h^3}{12\mu} \left.\frac{\partial P}{\partial y}\right|_L + \frac{C_p P(\theta, L)h_p^3}{12\mu w_p} = \frac{C_p P_{out}h_p^3}{12\mu w_p}.$$ (18)

The modeling of the inlet and outlet boundary conditions proposed in this work is simplified with respect to what can be found in other references [35,39]. The future improvement of the proposed model will re-evaluate the modeling of these boundary conditions, especially if different configurations of sealing and feeding mechanisms are considered.

### 2.9. Circumferential Periodicity

The circumferential periodicity must be satisfied at the edges of the flattened geometry (Figure 3). To maintain the continuity, the pressure and the circumferential gradient of the pressure along the axial direction must be equal on both sides. It was noted that the assignment of the pressure boundary condition is enough for the circumferential periodicity: the pressure gradient calculated on the two edges is equal.

### 2.10. Forces and Force Coefficients

Once the geometry and the mesh are defined and the boundary conditions are assigned, the Reynolds equation can be solved, and the pressure distribution obtained. The forces acting on the journal can be obtained by integrating the pressure profile along the circumferential and axial directions as follows:

$$\begin{bmatrix} F_x \\ F_y \end{bmatrix} = -\int_0^L \int_0^{2\pi} P(\theta, z, t) \begin{bmatrix} cos \\ sin\theta \end{bmatrix} R \, d\theta dy.$$ (19)

where $F_x$ and $F_y$ are the horizontal and vertical force in the absolute reference frame respectively (Figure 2).

The dynamic behavior of the SFD is represented by the force coefficients. As reported in many sources, among them [6,22], the SFD itself does not generate any kind of stiffness because, without the journal spinning, no pressure is generated at a given static displacement if there is no precession. The SFD forces are represented in the linearized form as follows:

$$\begin{bmatrix} F_x \\ F_y \end{bmatrix} = -\begin{bmatrix} C_{xx} & C_{xy} \\ C_{yx} & C_{yy} \end{bmatrix} \begin{bmatrix} v_x \\ v_y \end{bmatrix} - \begin{bmatrix} M_{xx} & M_{xy} \\ M_{yx} & M_{yy} \end{bmatrix} \begin{bmatrix} a_x \\ a_y \end{bmatrix},$$ (20)

where $v_x$ and $v_y$ are the instantaneous journal velocities and $a_x$, $a_y$ are the instantaneous journal accelerations.

Damping and added mass coefficients along the x and y directions are typical of small shaft orbiting around the static equilibrium position. In case of circular centered orbits, the SFD generates a constant reaction film force in a relative frame rotating with frequency $\omega$. In most rotodynamic applications, linearized force coefficients are considered. They represent changes in bearing reaction forces to infinitesimal amplitude motions about an equilibrium position. As the definition states, these coefficients are applicable only in the case of small motions around an equilibrium position. As reported in [40], in SFDs the orbit radius can go to half the clearance, defining an orbit far from being close to the equilibrium

position and thus violating the main hypothesis behind linearized force coefficients. For this reason, an orbit-based model, such as the one proposed in [40], is adopted in this work. The counterclockwise orbit of the SFD is divided into points where the forces are evaluated. The equation of motion is then written in the frequency domain by applying the Fourier transform to both the orbit points and the forces:

$$
\begin{bmatrix} F_x(\Omega) \\ F_y(\Omega) \end{bmatrix} = -\left( i\Omega \begin{bmatrix} C_{xx} & C_{xy} \\ C_{yx} & C_{yy} \end{bmatrix} - \Omega^2 \begin{bmatrix} M_{xx} & M_{xy} \\ M_{yx} & M_{yy} \end{bmatrix} \right) \begin{bmatrix} X(\Omega) \\ Y(\Omega) \end{bmatrix}. \tag{21}
$$

Equation (21) can be rewritten as follows:

$$
\begin{bmatrix} F_x(\Omega) \\ F_y(\Omega) \end{bmatrix} = -H(\Omega) \begin{bmatrix} X(\Omega) \\ Y(\Omega) \end{bmatrix}, \tag{22}
$$

where $H_{ij}$ coefficients are the four unknowns in Equation (22), but only two equations are available. For this reason, the same procedure is applied to the clockwise orbit, obtained by applying a negative value of $\omega$. So, the final system to be solved is

$$
\begin{bmatrix} F_x^{cc}(\Omega) \\ F_y^{cc}(\Omega) \\ F_x^{c}(\Omega) \\ F_y^{c}(\Omega) \end{bmatrix} = -H(\Omega) \begin{bmatrix} X^{cc}(\Omega) \\ Y^{cc}(\Omega) \\ X^{c}(\Omega) \\ Y^{c}(\Omega) \end{bmatrix}, \tag{23}
$$

where the apex $c$ stands for clockwise and the apex $cc$ stands for counterclockwise.

Once the matrix of complex stiffness $H$ is obtained, the single coefficients can be calculated as:

$$
C_{ij} = \frac{Imag(h_{ij})}{\omega}, \tag{24}
$$

$$
M_{ij} = -\frac{Real(h_{ij})}{\omega^2}. \tag{25}
$$

## 3. Model Validation

The model was validated with both numerical and experimental data available in the literature. The numerical and experimental results presented in [22] were considered due to the different geometrical configuration tested. In this work, four configurations (SFD A, B, E and F) were selected as reference for the validation. They differ in terms of clearance, SFD length, as well as the presence of a central groove and an end seal. The tested diameter is constant and equal to 127 mm. In [22], the oil has density $\rho_L = 805 \text{ kg/m}^3$ and dynamic viscosity $\mu_L = 0.0265$ Pa·s. The geometrical characteristics of the SFDs considered for the validation are listed in Table 1.

**Table 1.** Geometrical characteristics of SFDs from [22]. $d_G$ and $L_G$ represent the physical depth and length of the central groove, not present in SFD E and F. $d_E$ and $L_E$ represent the depth and length of the grooves at the discharge, not present in SFD E and F. Piston ring seals are applied only for SFD B.

|  |  | SFD A | SFD B | SFD E | SFD F |
|---|---|---|---|---|---|
| Clearance | $c_l$ [mm] | 0.141–0.251 | 0.138 | 0.122 | 0.267 |
| Length | $L$ [mm] | 2 × 25.4 | 2 × 12.7 | 25.4 | 25.4 |
| Central groove depth | $d_G$ [mm] | 9.5 | 9.5 | no | no |
| Central groove length | $L_G$ [mm] | 12.5 | 12.5 | no | no |
| End groove depth | $d_E$ [mm] | 3.5 | 3.5 | no | no |
| End groove length | $L_E$ [mm] | 2.5 | 2.5 | no | no |
| Seal | - |  | yes | yes | no | no |

SFDs E and F are tested with a fixed static eccentricity and by changing the amplitude of the circular orbit vibration. The considered $e/cl$ ratios are: 0.05, 0.14, 0.29, and 0.43.

The tested frequencies are $10 \div 250$ Hz for SFD E and $10 \div 100$ Hz for SFD F. The obtained coefficients are constant for the whole frequency range, therefore only the values at 100 Hz and 50 Hz are shown respectively. The evolution of both the mass and damping coefficients for SFD F is shown in Figure 7, where it is possible to see that the results obtained with the model presented in this paper agree well with both the experimental and numerical results in [22].

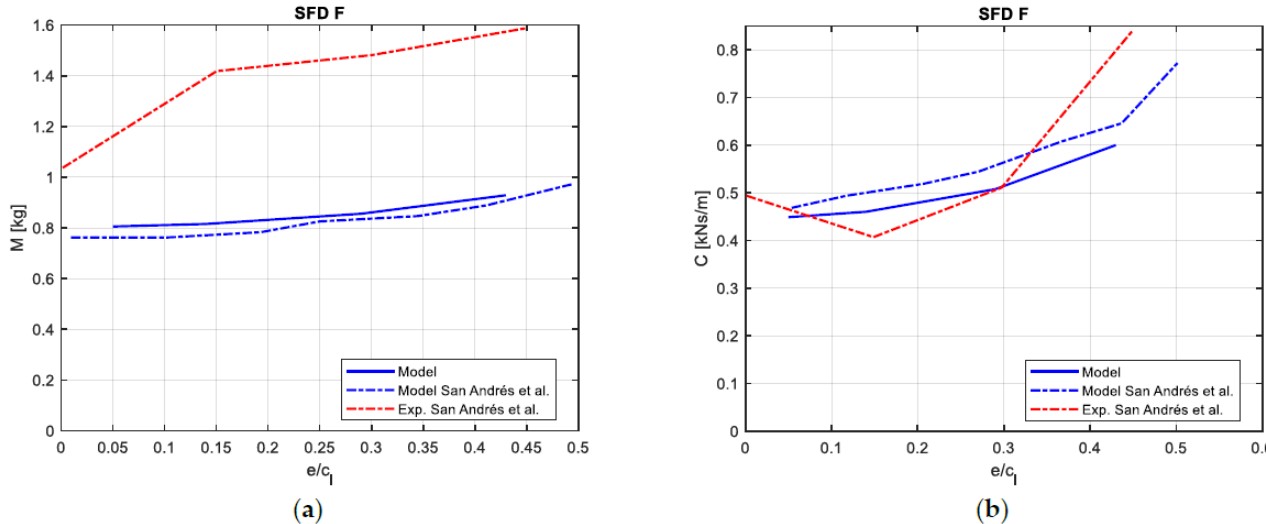

**Figure 7.** SFD F: numerical and experimental results from [22]. (**a**) Evolution of mass coefficient as a function of the orbit radius; (**b**) Evolution of damping coefficient as a function of the orbit radius.

Both the numerical results for the mass coefficient shown in Figure 7a underestimate the experimental results. In [22], the authors attribute the discrepancy to the high value of the feeding pressure that determines a higher value of the radial component of the force, directly responsible for the mass coefficient.

Similarly, the evolution of the mass and damping coefficient for SFD E is shown in Figure 8. In this case only the experimental results are available. It is possible to notice an acceptable agreement for the damping coefficients, the maximum difference between the experimental and numerical results is lower than the 25%. On the other hand, an important discrepancy between for the mass coefficients is shown. A possible explanation could be the high level of the feeding pressure that strongly affects the dynamic pressure distribution.

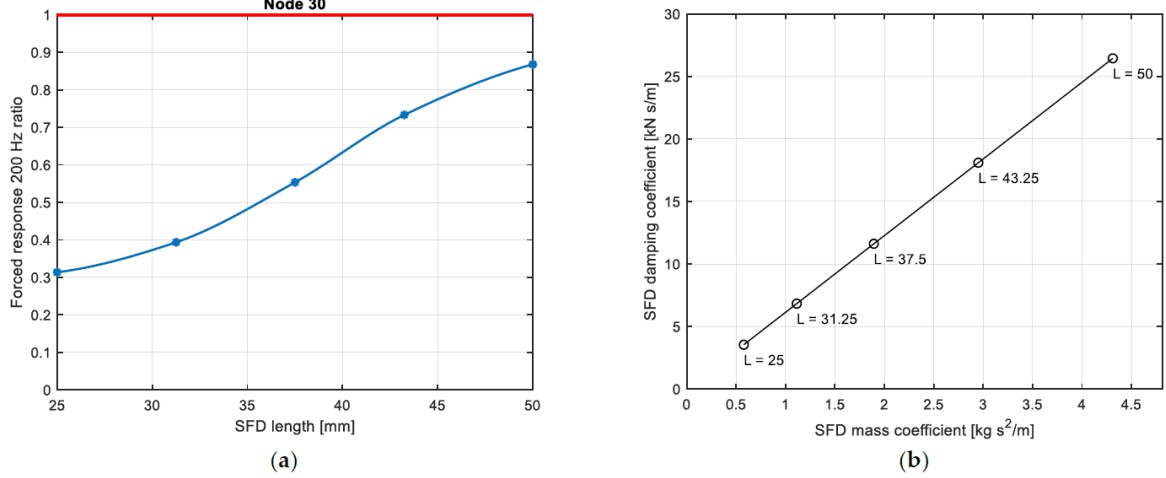

**Figure 8.** SFD E: experimental results from [22]. (**a**) Evolution of mass coefficient with orbit radius; (**b**) Evolution of damping coefficient with orbit radius.

SFDs B and A are tested in [22] at different static eccentricities with a constant orbit radius $e = 0.055c_l$ and for the frequency range $110 - 210$ Hz. Moreover, for these configurations, there is no variation of the force coefficients with the frequency and the only frequency considered is 150 Hz. For both configurations, the effective groove depth is tuned to match the results presented in [22]. The evolution of the dynamic coefficients with the static eccentricity for the open-ends configuration of SFD B is shown in Figure 9. Similarly to SFD F, the numerical results agree well with the experimental ones. A similar trend was obtained for SFD A. The results are not reported for the sake of brevity. In Figure 9, the values of the force coefficients are adimensionalized considering the same reference values reported in [22].

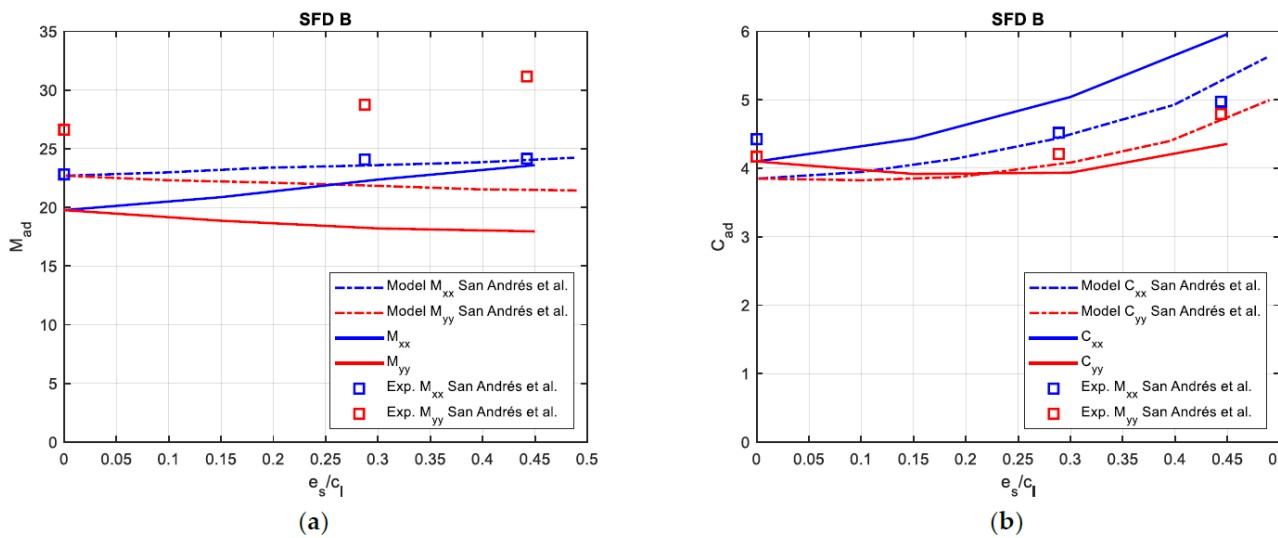

**Figure 9.** SFD B open ends: experimental and numerical results from [22]. (**a**) Evolution of mass coefficient with static eccentricity; (**b**) Evolution of damping coefficient with static eccentricity.

For every SFD configuration, the mesh independency check was performed considering a structured grid. Both rectangular and squared grids were considered, and the final number of elements adopted was selected with a trade-off between numerical accuracy and computational time. The evolution of the radial and tangential force relative error with the number of mesh points is reported.

For the sake of brevity, only the evaluation conducted for SFD F is reported. The number of axial points $N_z$ is selected and the tangential point $N_x$ are evaluated as $N_x = kN_z\frac{2\pi R}{L}$ with $0 < k \leq 1$. When $k = 1$, the elements are squared.

The evolution of the radial and tangential forces as a function of the number of mesh points and for some values of $k$ (0.05, 0.125 0.5, 1) is shown in Figure 10. Increasing the number of mesh points both errors reach an asymptote. When $k$ is reduced, i.e., when for the same number of axial points, the number of tangential points is reduced, the shape of the error evolution is flat. Generally, a relative error below 1% can be considered acceptable. To keep the alculation time low, for SFD F, the mesh configuration selected has $k = 0.125$ and approximately $1 \times 10^4$ mesh points.

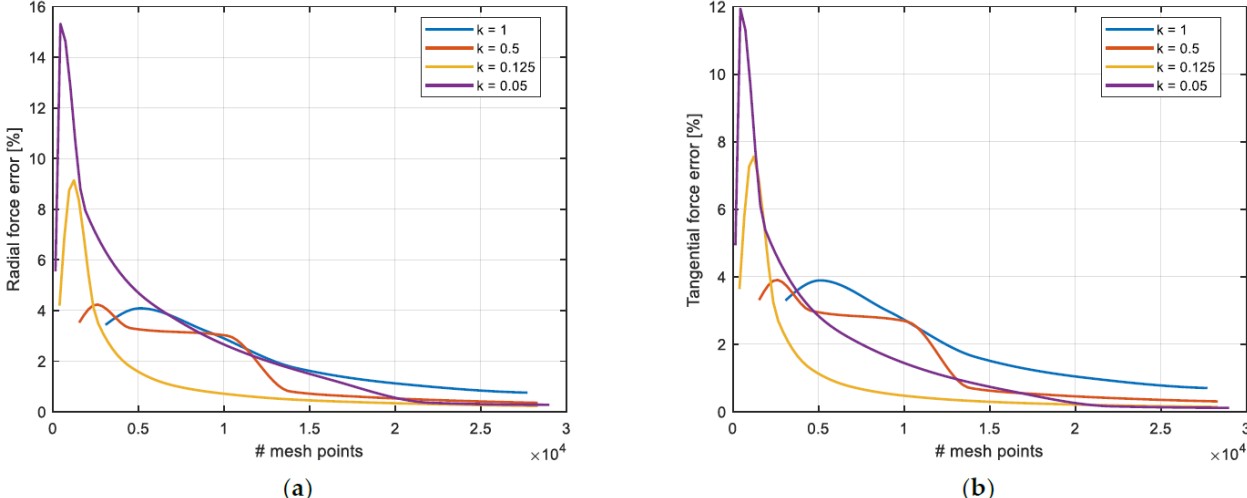

**Figure 10.** (**a**) Radial force error evolution with number of mesh points for SFD F; (**b**) tangential force evolution with number of mesh points for SFD F.

## 4. Application

The proposed model has been integrated in the finite beam element analysis of a high-speed centrifugal compressor coupled with a gear element. The shaft of the machine is long 0.7 m and the nominal diameter is 50 mm. The impeller is 70 mm long and has a maximum diameter of 140 mm, while the minimum one is 33 mm. The finite element discretization of the structure, with a total of 34 nodes, is shown in Figure 11.

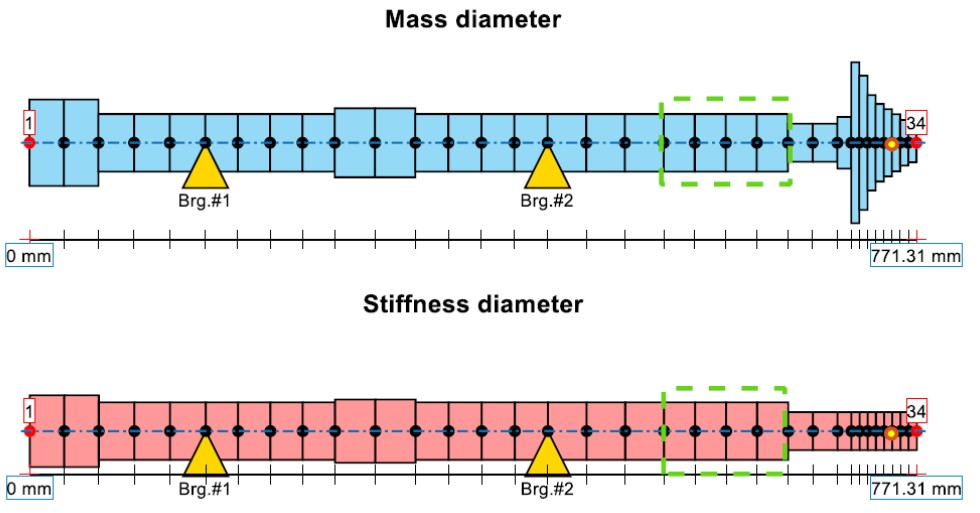

**Figure 11.** Finite element discretization of the machine from node 1 to node 34, sealing element in green rectangle.

As shown in Figure 11, the stiffness and mass diameter are different for the different elements. The yellow triangles represent the two roller element bearings. The green rectangle represents the region where a sealing element is placed. The scheme of the machine represents an actual application while the application investigated in the next pages is a hypothesis. In the analysis an unbalance force of $3 \times 10^{-6}$ [Kg·m] is placed in the yellow node of the impeller (node 31). The effect of the seal is not taken into consideration while the attention is focused on the reduction of the vibration of the machine, focusing on the impeller. The operational speed range of the compressor goes from $0 - 300$ Hz and 200 Hz is considered as the operating frequency. The forced responses to the unbalance at three nodes of the impeller are shown in Figure 12. It is possible to see that, due to the characteristics of the bearings, the system is barely damped and when crossing the natural

frequency, at 186 Hz, the vibration's amplitude is, in the last node, higher than $2 \times 10^{-4}$ m. Due to the small gaps between the impeller and the cage and to reduce the aerodynamic losses, it is important to reduce as much as possible the level of the vibration.

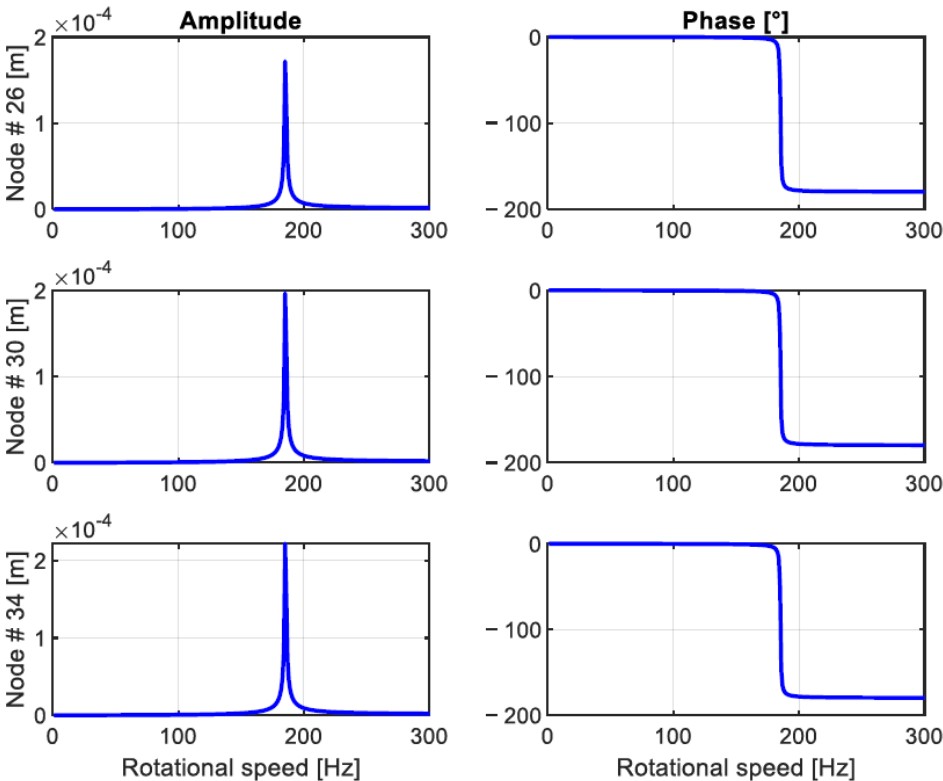

**Figure 12.** Amplitude and phase of vibration at nodes 26, 30, and 34.

To reduce the vibration peak, a SFD is applied in parallel with the first bearing. The new structure is shown in Figure 13. The SFD is supposed to be supported by an external squirrel cage defined by its own mass ($m_{cage}$) and stiffness ($k_{cage}$), respectively. Moreover, the squirrel cage acts as a centering mechanism. The SFD introduces an external source of damping ($c_{SFD}$) and added mass ($m_{SFD}$).

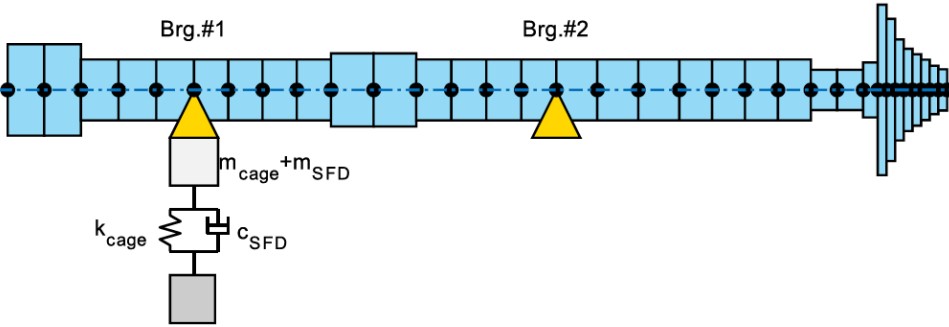

**Figure 13.** Finite element discretization with SFD.

For simplicity, a plain SFD without grooves, feeding system, and seals is considered. The geometrical characteristics of the SFD and the properties of the ISO VG 46 oil considered are listed in Table 2.

**Table 2.** Geometrical characteristics and oil properties of the SFD.

|  |  | SFD |
|---|---|---|
| Clearance | $c_l$ [mm] | 0.3 |
| Length | $L$ [mm] | 30 |
| Diameter | $D$ [mm] | 100 |
| Oil dynamic viscosity | $\mu_L$ [Pa·s] | 0.0775 |
| Oil density | $\rho_L$ [Kg/m$^3$] | 870 |
| Cage mass | $m_{cage}$ [kg] | 1.16 |
| Cage stiffness | $k_{cage}$ [N/m] | $2 \times 10^7$ |

At first, the forced response of the configuration with the squirrel cage but without considering the presence of the oil is performed. The comparison between the two forced responses for the same impeller nodes considered in Figure 12 is shown in Figure 14.

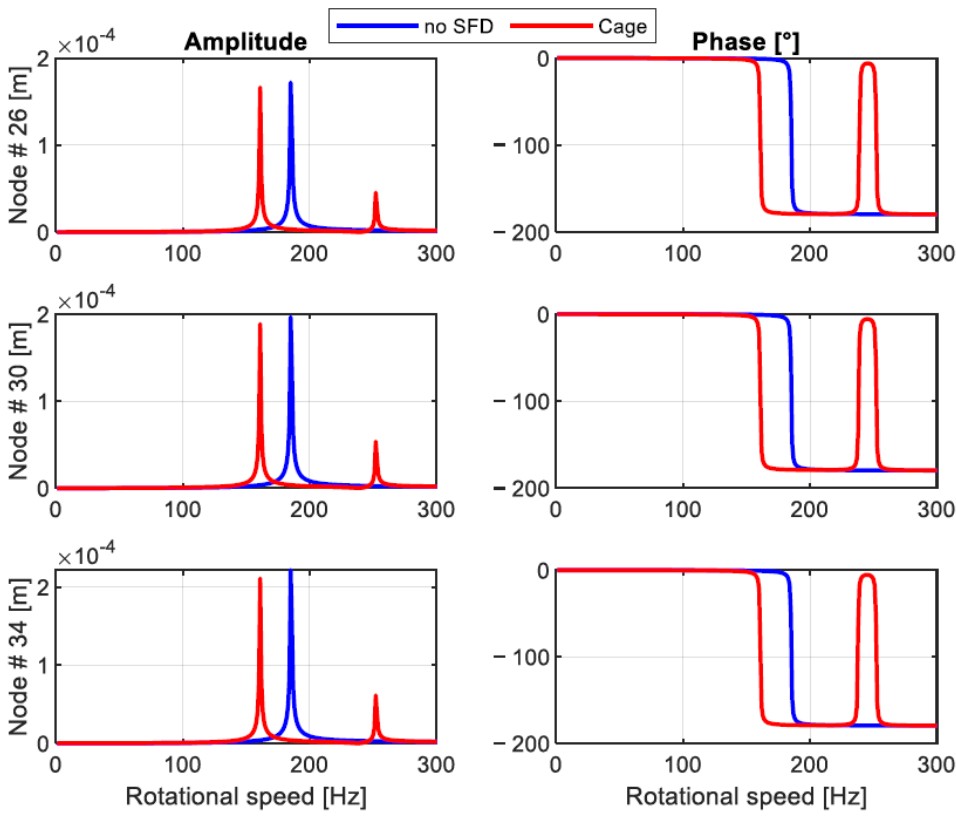

**Figure 14.** Forced response comparison for the original configuration and the configuration with the squirrel cage.

As it is possible to see from Figure 14, the introduction of the squirrel cage significantly changes the forced response. The introduction of the squirrel cage has the effect of a tuned mass damper. The resonance peak at 182 Hz is moved to 161 Hz. Moreover, a second resonance peak is present at 252 Hz.

Then, the previously mentioned SFD is considered for the forced response. The comparison between the forced response between the original configuration, the configuration with the squirrel cage, and the SFD configuration is shown in Figure 15. The introduction of the SFD is strongly effective in the reduction of the level of the vibration peak.

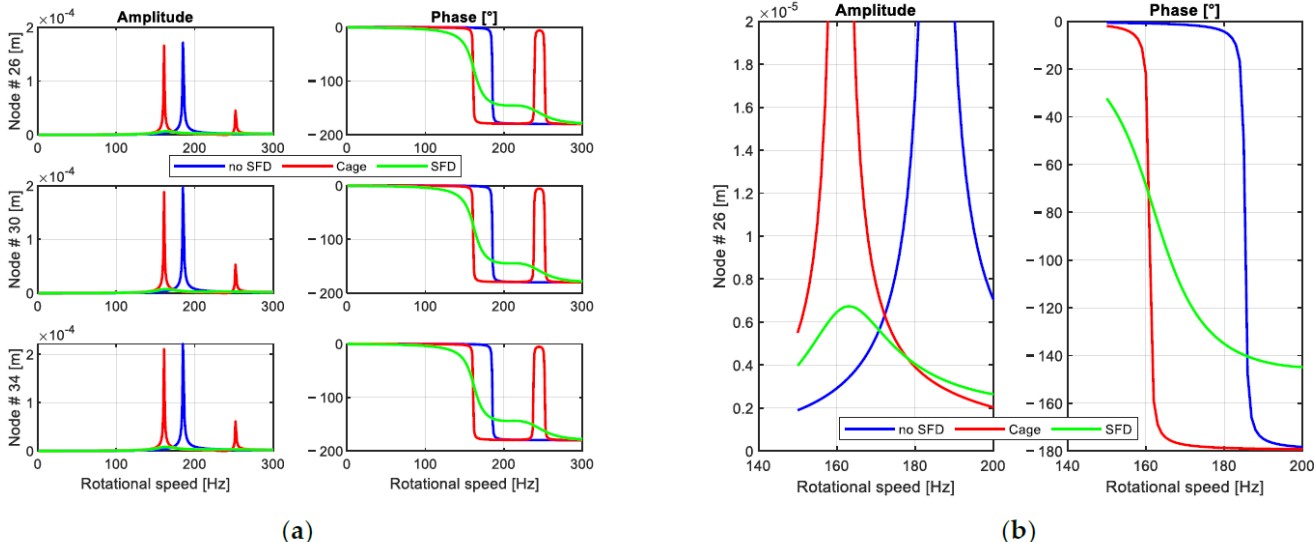

**Figure 15.** Forced response comparison for the original configuration, the configuration with the squirrel cage, and the configuration with the SFD (**a**). Forced response comparison for the original configuration, the configuration with the squirrel cage, and the configuration with the SFD for node 34 (**b**).

Then, the effect of some geometrical parameters on the forced response of the system is evaluated. As previously mentioned, the operating frequency considered is 200 Hz. Therefore, considering the evolution of the forced responses shown in Figure 15, the frequency range from 150 Hz to 200 Hz is considered for the following analysis.

### 4.1. SFD Clearance

The first parameter to be investigated is the clearance of the SFD. The ratio between the forced response at 200 Hz of the configuration with the SFD and the original one is shown in Figure 16. The forced response decreases with the increase of the SFD clearance even though the damping coefficients increases when the SFD clearance is decreased. This behavior is related to the increase of the resonance frequency when the SFD clearance is reduced. Therefore, a higher level of vibration is obtained at 200 Hz (see Figure 17).

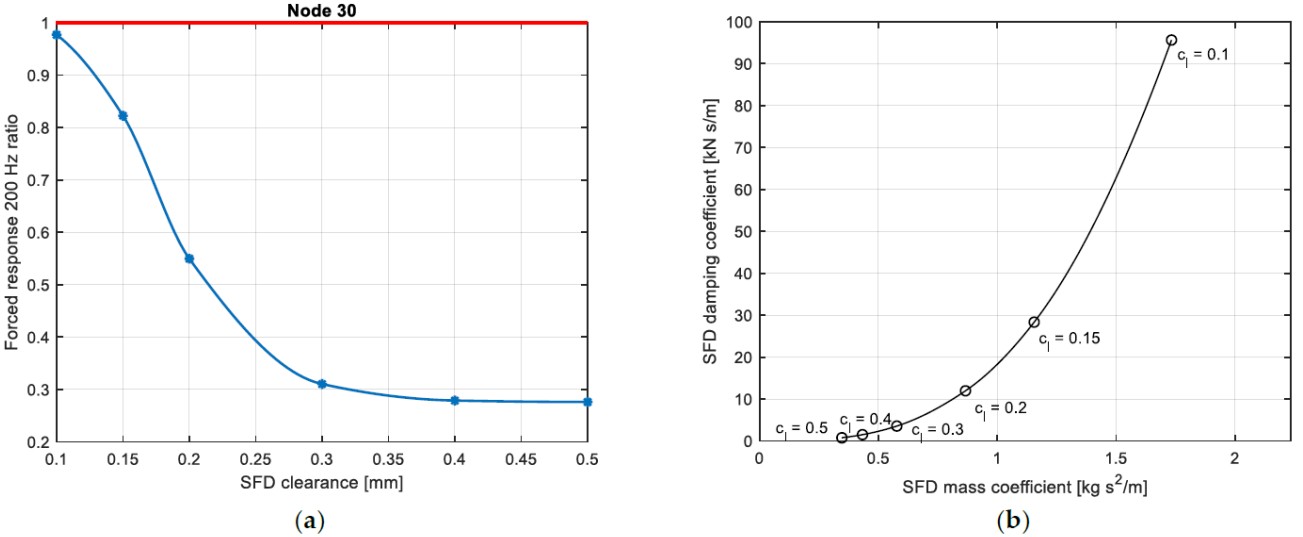

**Figure 16.** Forced response ratio at 200 Hz for node 30 for different values of SFD clearance (**a**), evolution of SFD force coefficients with clearance (**b**).

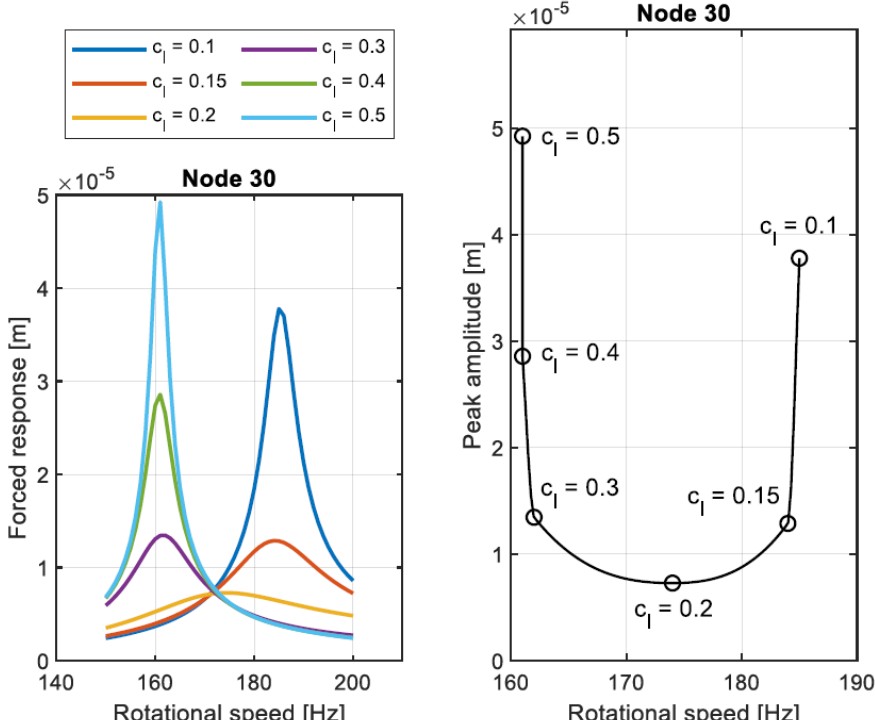

**Figure 17.** Comparison of forced response for different values of SFD clearance, circle markers indicate the clearance value.

### 4.2. SFD Length

The effect of the length of the SFD on the forced response is investigated. For this analysis, the selected SFD clearance is 0.3 mm because it minimizes the vibration level at 200 Hz and guarantees an acceptable level for the resonance peak vibration. The ratio of the forced response at 200 Hz for the original configuration and the configuration with the SFD is shown in Figure 18a. The minimum forced response is obtained when the shortest SFD is considered. The force coefficients of the SFD increase with the SFD length, see Figure 18b. Therefore, also in this case, the most suitable SFD to reduce the vibration level at 200 Hz is the one characterized by the lowest force coefficients.

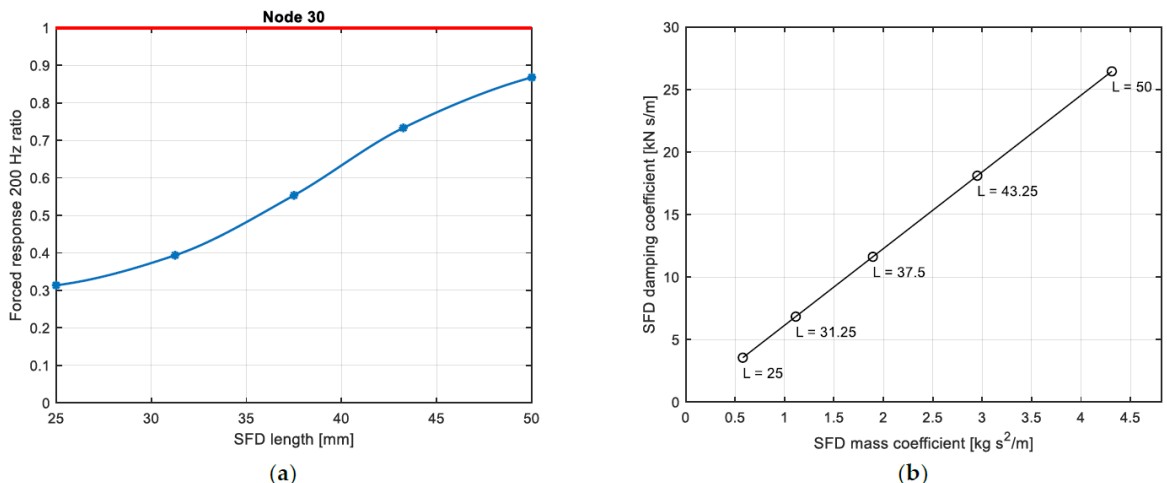

**Figure 18.** Forced response ratio at 200 Hz for node 30 for different values of SFD length (**a**), evolution of SFD force coefficients with length (**b**). Red line for system without SFD, blue line indicates system with SFD of (**a**). Markers highlights tested configurations.

The comparison between the forced responses obtained considering the different values of length of SFD is shown in Figure 19. Moreover, in this case, the results are reported in the frequency range of interest. It is possible to see that the minimum forced response at 200 Hz is obtained with the shortest damper. On the contrary, the minimum of the vibration peak is obtained when $L = 37.5$ mm, as shown in the right part of Figure 19. Therefore, the proper SFD configuration must be selected according to the optimization required.

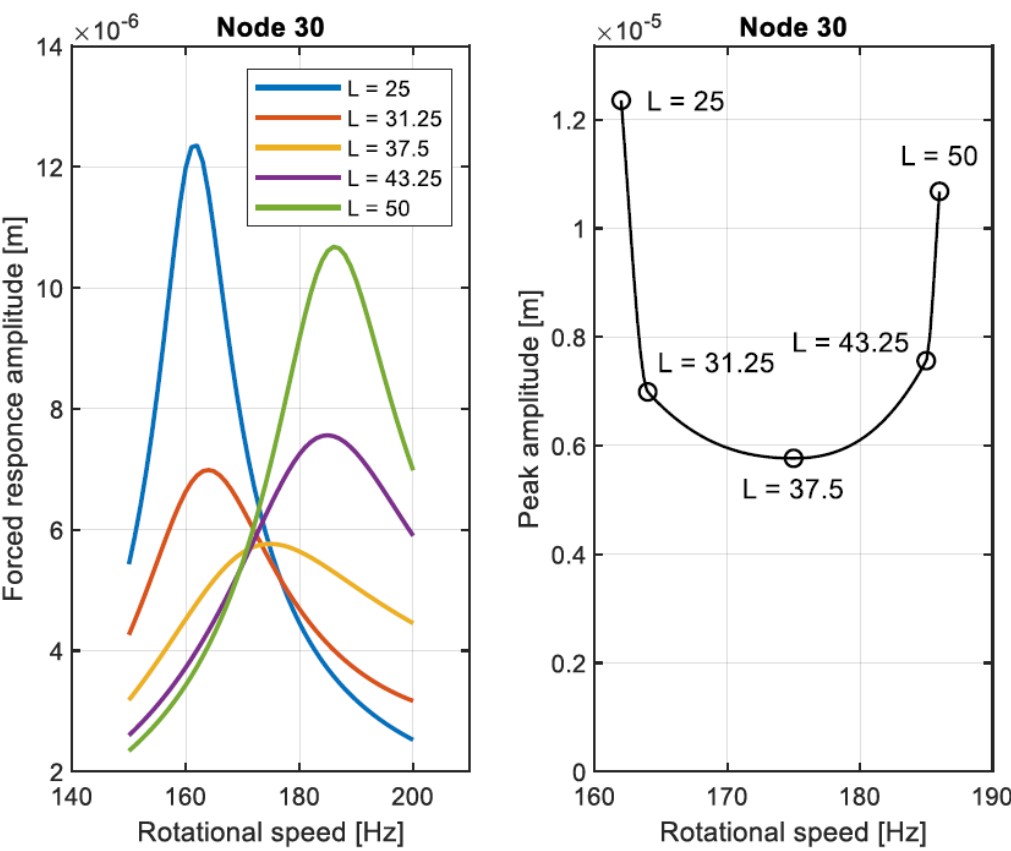

**Figure 19.** Comparison of forced response for different values of SFD length at node 30, markers highlight configurations tested.

### 4.3. Groove Effective Depth

Another tuning parameter that can be selected for the geometry of the SFD is the effective depth of the groove. For this reason, different values of $d_{ge}$ have been investigated. For convenience, the relative value of the effective groove depth is considered ($d_{ge}/cl$). The ratio of the forced response at 200 Hz for the original configuration and the configuration with the grooved SFDs is shown in Figure 20a. The clearance considered is 0.3 mm and the lands of the SFD have a length of 12.5 mm. The groove length considered is 5mm and it is placed in the center of the SFD. When the effective groove depth is one, the damper geometry results in a grooveless damper of length 30 mm. From the analysis shown in Figure 20a, the higher the groove depth, the lower the forced response at 200 Hz. The evolution of the ratio between the SFD force coefficients for the different values of the relative effective groove depth and the values obtained when $d_{ge}/cl = 1$ is shown in Figure 20b. Increasing the groove depth, the damping coefficient is reduced while the mass coefficient is highly increased. The evolution of the forced responses for the considered frequency range is shown in Figure 21. Moreover, in this case, the configuration that minimizes the forced response at 200 Hz is not the one that minimizes the amplitude of the peak.

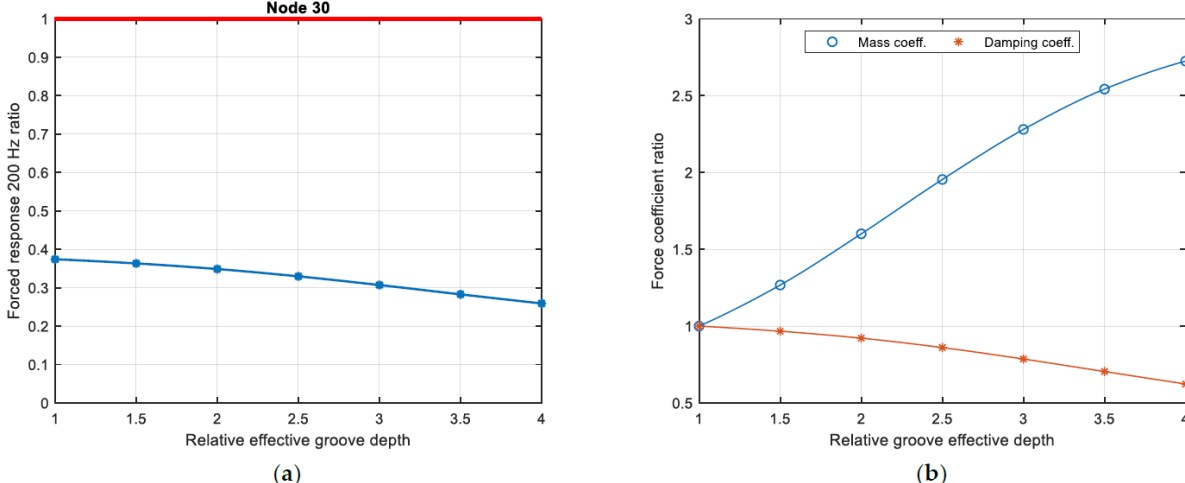

**Figure 20.** Forced response ratio at 200 Hz for node 30 for different values of relative effective groove depth (**a**), evolution of SFD force coefficients ratio with effective groove depth (**b**). Red line for system without SFD, blue line indicates system with SFD of (**a**). Markers highlights tested configurations.

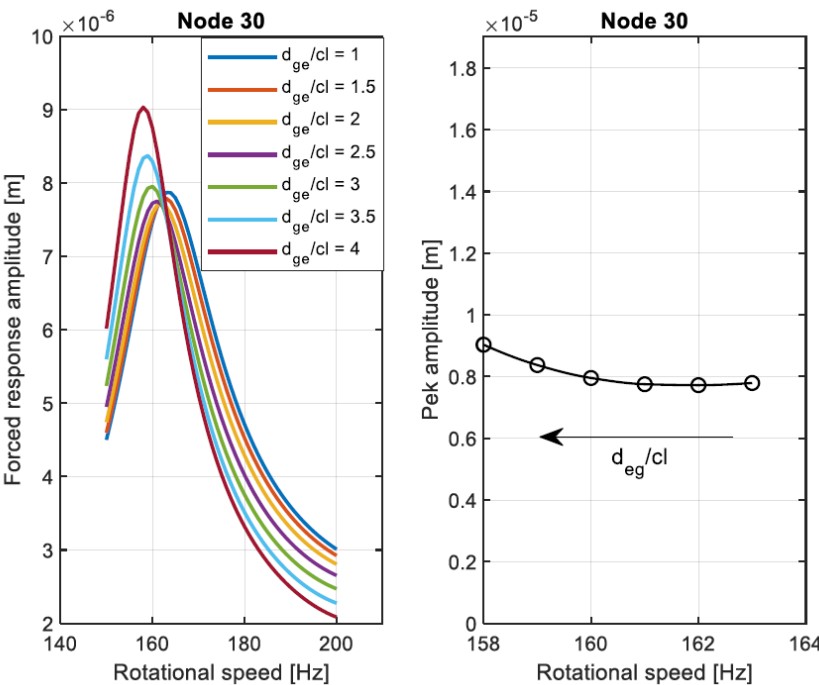

**Figure 21.** Comparison of forced response for different values of relative groove effective depth at node 30. Markers highlight tested configurations, arrow highlights growing direction of $d_{eg}/cl$.

### 4.4. Feeding Pressure

In this section, the feeding system is considered. The SFD considered has a length of 30 mm and clearance equal to 0.3 mm, and the diameter of the holes is considered equal to 2 mm. Moreover, the coefficient $C_i$ is considered as $1 \times 10^{-9} \ \frac{m^3}{s \ Pa}$. The effect of the feeding pressure on the forced response has been investigated. The ratio of the forced response at 200 Hz for the original configuration and the configuration with SFDs is shown in Figure 22. When the feeding pressure is considered zero, the feeding system is not included in the modeling. The presence of the feeding system seems to have a small impact on the forced response at 200 Hz, and the inlet pressure is not influencing the forced response.

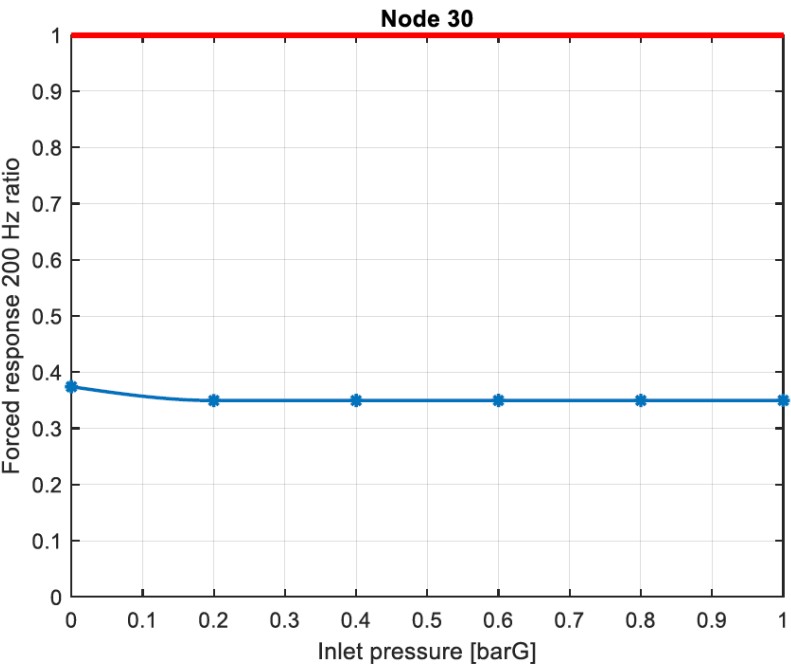

**Figure 22.** Forced response ratio at 200 Hz for node 30 for different values of feeding pressure depth. Red line for system without SFD, blue line indicates system with SFD.

If the feeding system is not considered in the modeling and no cavitation or air ingestion is present, both the direct SFD force coefficients are equal and constant with the tested frequencies. On the contrary, when the feeding system is modeled, the xx and yy force coefficients are slightly different. Moreover, the mass coefficients show a dependency with the frequency which is more evident when the feeding pressure is increased, as shown in Figure 23.

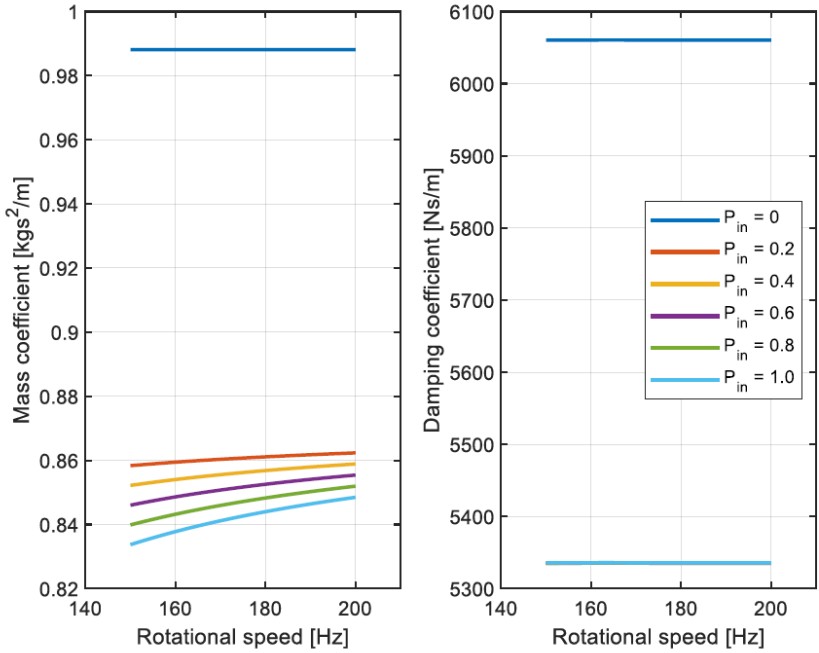

**Figure 23.** Evolution of xx mass and damping coefficient with the rotational speed considering different inlet pressure values.

The differences in the force coefficients shown in Figure 23 determine different forced response between the modeling with and without the feeding system (Figure 24). On the

contrary, the evolution of the mass coefficient with the rotational speed does not have an impact on the forced response.

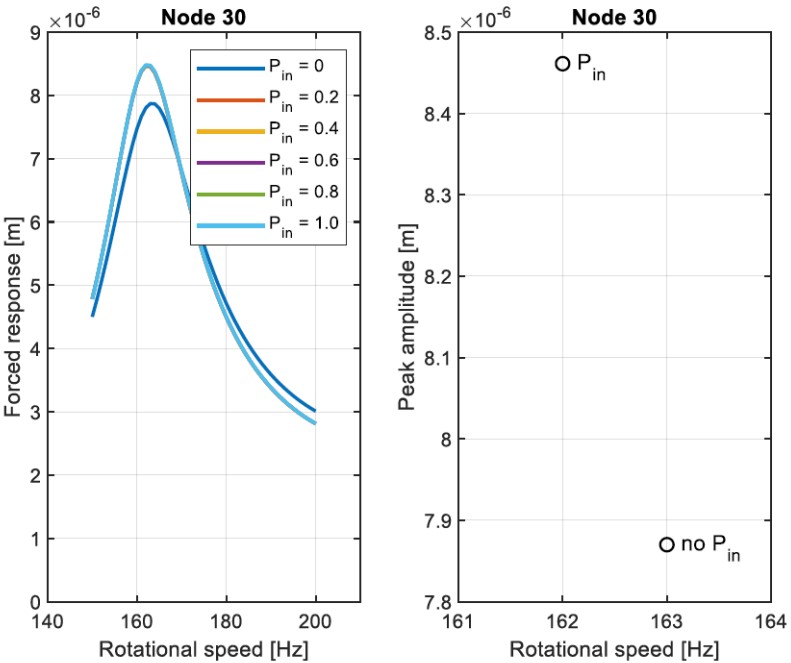

**Figure 24.** Comparison of forced response for different values of inlet pressure at node 30.

*4.5. End Seals*

For convenience, the effect of the sealing mechanism is shown considering the overall seal coefficient $C_{seal} = C_p h_p^3 / w_p \mu$. The highest value of $C_{seal}$ corresponds to the open ends condition while the lowest value of $C_{seal}$ corresponds to the ideal condition of complete sealing. The ratio of the forced response at 200 Hz for the original configuration and the configuration with the sealed SFDs is shown in Figure 25a. Increasing the sealing effect determines an increase of the forced response at 200 Hz at node 30. The evolution of the force coefficients with $C_{seal}$ at 200 Hz is shown in Figure 25b. Increasing the sealing effect determines an increase of both the force coefficients of the SFD.

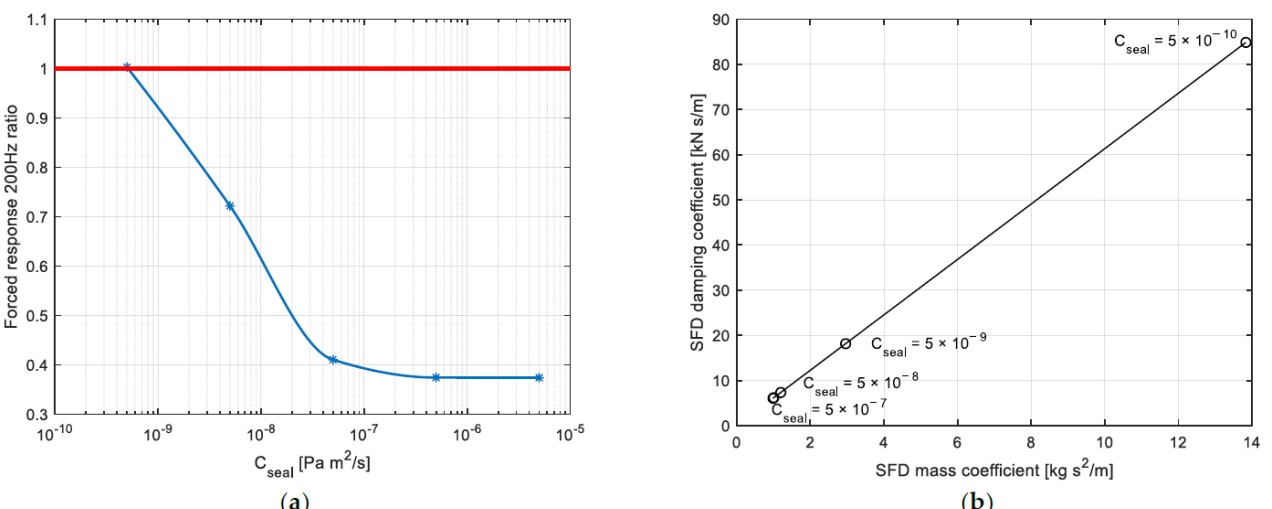

(a)                                                                                      (b)

**Figure 25.** Forced response ratio at 200 Hz for node 30 for different values of $C_{seal}$ (**a**), evolution of SFD force coefficients for different values of $C_{seal}$ (**b**). Red line for system without SFD, blue line indicates system with SFD of (**a**). Markers highlights tested configurations.

The evolution of the forced responses for the considered frequency range is shown in Figure 26. Moreover, in this case, when the force coefficients are increased, the frequency of the peak of the forced response increases.

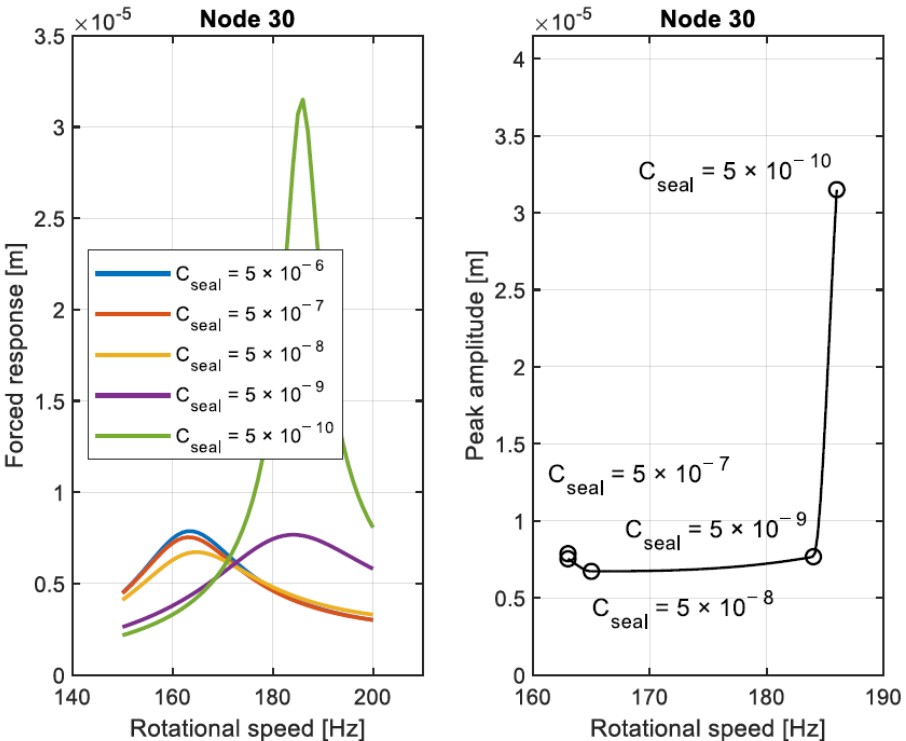

**Figure 26.** Comparison of forced response for different values of $C_{seal}$ at node 30.

### 4.6. Correction of an Instability

In this section the effect of the seal placed before the impeller is considered as source of instability. The stiffness matrix at the nodes of the seal is introduced as follows:

$$K_{seal} = \begin{bmatrix} 0 & k_{seal} \\ -k_{seal} & 0 \end{bmatrix}, \tag{26}$$

The parameter $k_{seal}$ determines whether the compressor is affected by instability. The effect of $k_{seal}$ on the stability of the system is investigated. The first instability is present at $k_{seal} = 15,000 \, \text{N/m}$. However, the system is unstable for the whole frequency range only when $k_{seal} \geq 17,500 \, \text{N/m}$. For this reason, our analysis is focused at $k_{seal} = 17,500 \, \text{N/m}$.

The same architecture shown in Figure 13 is considered. The damping introduced in the system by the SFD tends to have a stabilizing effect. The dimensionless damping factor is studied as an indication of the stabilizing effect. This indicator is defined as:

$$\eta_i = -\frac{Real(\lambda_i)}{Imag(\lambda_i)}, \tag{27}$$

The SFD considered is similar to that described in Table 2 but now the clearance is set to 0.5 mm. The dimensionless damping factor for the original system, system with the cage, and the system with the SFD is shown in Figure 27. Both the original system and the system with the cage are affected by instability. On the contrary, when the SFD is added to the system the correction of the instability is achieved.

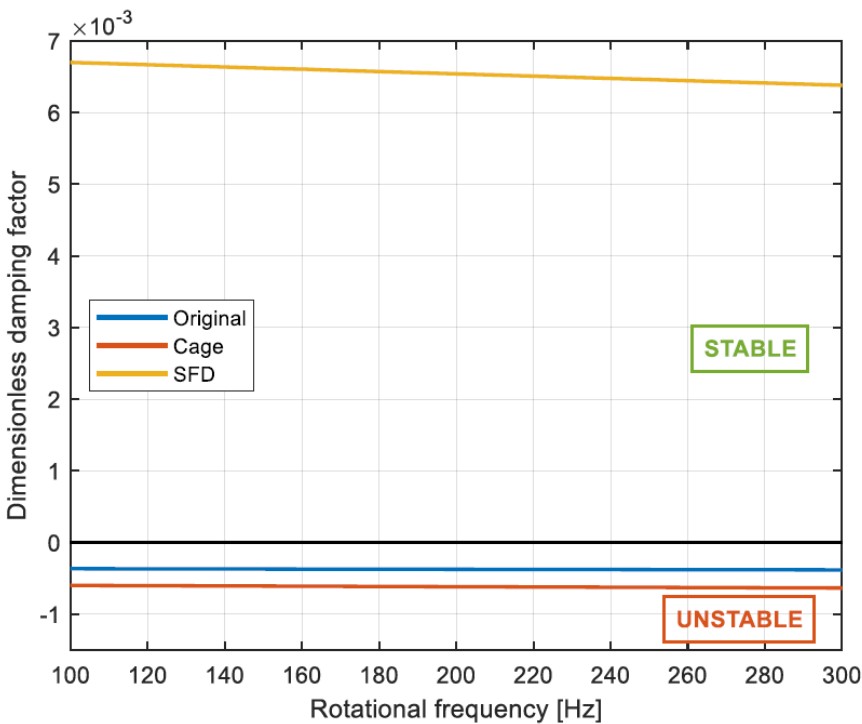

**Figure 27.** Evolution of dimensionless damping factor for unstable modes with rotational frequency of original system, system with cage, and system with SFD.

## 5. Conclusions

In this paper, a thorough investigation on the state of the art of SFDs is reported. The most critical features are highlighted and an evaluation of how to treat them is given. A comprehensive model based on the 2D Reynolds equation is presented. The classic Reynolds equation is modified to include an extra term to take into consideration the effect of the temporal inertia and the air ingestion modeling is also considered. The equation is numerically solved with the finite difference approach.

The proposed model is then validated with numerical and experimental results available in the literature. At first, the effect of the air ingestion is considered. Secondly, the overall effect of different geometrical configurations is considered. The results obtained show an acceptable agreement for the evaluated configurations.

A finite element code to simulate the dynamic behavior of turbomachines was developed. The effect of the SFD is included considering the force coefficients calculated from the finite difference solution of the Reynolds equation. A slightly unbalanced centrifugal compressor was considered and a parametric investigation on several parameters of the SFD was performed to test the effectiveness of the SFD in the vibration reduction. In general, the application of the SFD is effective in reducing the level of the vibration. From the investigation, it is highlighted that the selection of the most appropriate design of the SFD is highly influenced by the application. Moreover, the application of the SFD also proved effective in the correction of the instability.

The model derived in this paper has proven effective and efficient in the prediction of the dynamic properties of SFDs and can therefore be considered a useful tool in the initial design of these critical components. More accurate and precise models based on CFD simulations are present in the literature, but these are characterized by a higher level of complexity and longer simulation times.

**Author Contributions:** Conceptualization, E.G.; methodology, E.G. and S.C.; software, E.G. and S.C.; validation, E.G.; formal analysis, E.G.; investigation, E.G.; resources, P.P.; data curation, E.G.; writing—original draft preparation, E.G.; writing—review and editing, A.V.; visualization, E.G.; supervision, A.V.; project administration, P.P.; funding acquisition, P.P. All authors have read and agreed to the published version of the manuscript.

**Funding:** This research received no external funding.

**Data Availability Statement:** Data used are confidential and cannot be shared.

**Conflicts of Interest:** The authors declare no conflict of interest.

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
