# Peer review of "Squeeze Film Damper Modeling: A Comprehensive Approach"

_machines, doi:10.3390/machines10090781_

Round 1

Reviewer 1 Report

This manuscript introduces a numerical model based on the Reynolds equation, discretized with the finite difference method. Different boundary conditions for oil feeding and discharging are implemented and investigated. The manuscript were well written and the results were valuable for practical engineering. There are several issues should be addressed:

1. In Abstract, the author stated that the finite difference method was used to solve Reynolds equation, whereas it became finite element in Conclusion, please clarify.

2. In Fig.7, where the data plots coms from? Did they obtained from simulation results. The author should clarify.

3. The validation of present simulation should also be given through mesh independence test, such as the following reference. DOI: 10.1016/j.ijmecsci.2022.107468

4. The proposed model are based on the 2D Reynolds equation. Does it validate for 3D configuration, since the practical engineering is 3D configuration.

5. More recent publications should be added in Reference and Introduction.

6. The unit should written in standardized form. The variables should be italic, please check.

Author Response

See attached file for the replies to the comments

Reviewer 2 Report

This article concerns an interesting topic: characterization and modelling of squeeze film dampers considering the most relevant phenomena involved in their operation. The authors propose a simplified formulation which is validated with numerical and experimental results available in the literature. They also present an example of application of one of these dampers to a compressor rotor.

The equations used are already well-known or available in the literature, and the results obtained with the proposed model do not seem to improve those presented in other references. Therefore, I appreciate that the main contribution of the article is the illustrative application to a particular compressor rotor and the corresponding process of analysis of the effect of some design parameters on its performance.

The article is clear and well-written. On this occasion, I only have some minor remarks:

1. ? is defined as the “viscosity”. It would be more precise to indicate that ? is “dynamic viscosity”.

2. The use of the notation “cl“ for the clearance can cause misunderstanding because “cl“ seems to stand for a product of two different parameters. It would be clearer to use the usual symbol “c”, or alternatively “cl”.

3. The allusion to reference [19] (instead of [20]) in the legends of Figures 7-9 seems erroneous.

Author Response

See the attached file for the review to the comments

Round 2

Reviewer 1 Report

It can be accepted.